# Intranasal Methylprednisolone Ameliorates Neuroinflammation Induced by Chronic Toluene Exposure

**DOI:** 10.3390/pharmaceutics14061195

**Published:** 2022-06-02

**Authors:** Manuel F. Giraldo-Velásquez, Iván N. Pérez-Osorio, Alejandro Espinosa-Cerón, Brandon M. Bárcena, Arturo Calderón-Gallegos, Gladis Fragoso, Mónica Torres-Ramos, Nayeli Páez-Martínez, Edda Sciutto

**Affiliations:** 1Departamento de Inmunología, Instituto de Investigaciones Biomédicas, Universidad Nacional Autónoma de México, Avenida Universidad 3000, Ciudad Universitaria, Coyoacán, Ciudad de México 04510, Mexico; manuel_f_er@hotmail.com (M.F.G.-V.); nicolas2309@outlook.es (I.N.P.-O.); alexec2803@gmail.com (A.E.-C.); bmbc1520@gmail.com (B.M.B.); arturo.calderon@comunidad.unam.mx (A.C.-G.); gladisfragoso@hotmail.com (G.F.); 2Unidad Periférica de Neurociencias, Facultad de Medicina UNAM-Instituto Nacional de Neurología y Neurocirugía, MVS-SSA, Insurgentes sur 3877, La Fama, Tlalpan, Ciudad de México 14269, Mexico; monica.torres@innn.edu.mx; 3Laboratorio Integrativo Para el Estudio de Sustancias Inhalables Adictivas, Dirección de Investigaciones en Neurociencias, Instituto Nacional de Psiquiatría Ramón de la Fuente Muñiz, Calzada México-Xochimilco 101, Tlalpan, Ciudad de México 14370, Mexico; 4Sección de Posgrado e Investigación, Escuela Superior de Medicina, Instituto Politécnico Nacional, Plan de San Luis y Díaz Mirón, Miguel Hidalgo, Ciudad de México 11340, Mexico

**Keywords:** toluene, neuroinflammation, histological damage, intranasal administration, methylprednisolone

## Abstract

Inhalants are chemical substances that induce intoxication, and toluene is the main component of them. Increasing evidence indicates that a dependence on inhalants involves a state of chronic stress associated to the activation of immune cells in the central nervous system and release of proinflammatory mediators, especially in some brain areas such as the nucleus accumbens and frontal cortex, where the circuits of pleasure and reward are. In this study, anti-neuroinflammatory treatment based on a single dose of intranasal methylprednisolone was assessed in a murine model of chronic toluene exposure. The levels of proinflammatory mediators, expression levels of Iba-1 and GFAP, and histological changes in the frontal cortex and nucleus accumbens were evaluated after the treatment. The chronic exposure to toluene significantly increased the levels of TNF-α, IL-6, and NO, the expression of GFAP, and induced histological alterations in mouse brains. The treatment with intranasally administered MP significantly reduced the expression of TNF-α and NO and the expression of GFAP (*p* < 0.05); additionally, it reversed the central histological damage. These results indicate that intranasally administered methylprednisolone could be considered as a treatment to reverse neuroinflammation and histological damages associated with the use of inhalants.

## 1. Introduction

Inhalant abuse is a worldwide problem especially common in youths from marginalized populations [1]. These substances are widely available and accessible to children and adolescents, in whom they have euphorigenic and highly toxic effects, which may be greater than in adults [2]. Toluene is the main substance contained in inhaled drugs [3,4], and its exposure is related to different neurological alterations [5,6]. The addiction within inhalant abuse underlies a neurochemical and neuroanatomical circuit of pleasure and reward. In this circuit, the nucleus accumbens, the hippocampus, the frontal cortex, and the amygdala are the most important brain areas. Of them, the nucleus accumbens is critical for the beginning and the maintenance of the reinforcement of the drug abuse behavior [7]. Considering these reasons, in this study, histological analysis was performed in mouse frontal cortices (FCs) as well as nucleus accumbens (NA).

At cellular levels, chronic toluene administration impact cell viability by decreasing neurogenesis and promoting apoptosis [6]. Moreover, chronic toluene exposure produces several cognitive deficits, including learning and working memory impairment [8,9]. Among other effects, toluene inhalation induces changes in GABA and glutamate receptors, although the mechanisms that underlie these effects remain to be determined [10,11,12,13].

Neuroinflammation is the complex inflammatory response of neural tissue that initially participates in the brain fix and resolution process [14] but has also been attributed to the pathogenesis of a number of central nervous system diseases [15]. Neuroinflammation may be induced by toluene inhalant abuse due to the production of reactive oxygen species and glial cell activation [16]. In this way, different studies have demonstrated that drugs of abuse, such as toluene, interact with the neuroimmune system and alter immune gene expression and signaling, which in turn contribute to various aspects of addiction [17]. It has been reported that toluene exposure enhances IL-1β mRNA while decreasing TGF-β mRNA expression in the medial prefrontal cortex [16]. In addition, low toluene concentrations in mice may increase the expression of mRNA of TLR-4 and NF-KB mRNA in the hippocampus [18]. Furthermore, an increase in the expression of cell-specific markers ionized calcium binding adaptor molecule (Iba-1) and glial fibrillary acidic protein (GFAP) has been reported following exposure to toluene [16,19]. It has also been reported that repeated toluene administration (1500 ppm for 4 h/7 days) increased the expression of cytokine mRNA such as TNF-α, TGF-β, and glial markers such as GFAP in the rat hippocampus and cerebellum [20]. In mice chronically exposed to toluene, rapid absorption occurs. Once absorbed, toluene can readily cross the blood–brain barrier, resulting in brain damage by triggering neuroinflammation through the release of damage-associated molecular patterns. 

Despite the worldwide use of toluene, the treatments for inhalant users are limited, and there is a clear need for more effective treatment options [21]. Some of these treatments include psychological and psychosocial interventions [22], and only a few studies have analyzed pharmacological approaches for the inhalant treatment [23]. In addition, in postmortem analysis of brains from patients undergoing inhalant addiction, it had been shown that gliosis, microglia migration and proliferation, cerebral atrophy, and myelin loss are the main effects of toluene exposure; it is important to say that for all the mentioned processes, neuroinflammation is critical for their establishment [24]. Altogether, these data suggest that neuroinflammation is a therapeutic target to reduce, at different levels, the harmful effects of toluene exposure.

Steroids are the drugs with the highest anti-inflammatory capacity and are commonly used in a wide range of pathologies such as allergies, asthma, neurodegenerative and autoimmune diseases, etc. [25,26]. Glucocorticoids are a class of steroids which regulate various cellular functions including, development, homeostasis, cognition, and the inflammatory response [27]. Despite the efficiency of glucocorticoids in controlling inflammation, high systemic doses are required to achieve central therapeutic doses required to control neuroinflammation, which may cause relevant adverse side effects [28]. Thus, its use is restricted to very particular neuroinflammatory conditions that endanger the patient’s life. 

In recent years, the intranasal (IN) route of administration has been recognized as a route to treat neurological disorders. The nasal pathway represents a non-invasive administration route of active pharmaceutical ingredients for local, systemic, and CNS action [29]. IN delivery not only avoids first-pass metabolism but also circumvents passage through the blood–brain barrier by allowing the transport of drugs from the nasal cavity to the brain through the nasal and trigeminal nerves [30]. On the other hand, the mucosa and lamina propria are characterized by their extensive vascularization; likewise, the leaky epithelium provides an optimal absorption mechanism for the drug delivery [31]. To our knowledge, there are no reports on the use of the IN drug administration route to control toluene-associated neuroinflammation. 

Considering that it has been reported that the repeated administration of different drugs of abuse in rodents causes a progressive increase in locomotor activity (Valjent et al., 2010) that is greater in magnitude compared to that induced by a single administration (a phenomenon known as locomotor sensitization), it is important to evaluate the impact of toluene inhalation and MP treatment on the addiction behavior. This activating effect of locomotion is easily perceptible and has a neurobiological substrate that is in part common with the one responsible for the reward effects. Thus, in this study, the impact of the anti-neuroinflammatory treatment in the addiction behavior was also evaluated. 

The aim of this study was to evaluate the efficiency of the IN delivery of methylprednisolone (MP) to control the neuroinflammation induced by the chronic exposure of toluene in mice and its impact on the addiction behavior. 

## 2. Materials and Methods

### 2.1. Mice

Male Swiss Webster mice of 30–35 days of age were employed in this study. Mice were obtained from Instituto Nacional de Psiquiatría Ramon de la Fuente Muñiz, México City, México, and housed at the same institution in an animal room maintained at 22 ± 3 °C with a 12/12 h light–dark cycle. The age of mice approximately corresponds to human adolescence, which is the stage of life at which people generally start drug addictions. All mice were kept in plexiglass boxes with food (Teklad Sani-Chips^®^ 7090, Envigo, Indianapolis, IN, USA) and water (filtered, acidified, and sterilized) ad libitum.

Experimental procedures were conducted following the guidelines of the Institutional Committee of the Care and Use of Experimental Animals of the IIB at the Universidad Nacional Autónoma de México (UNAM) and of the U.S. National Institutes of Health. Experimentation protocol was approved by the animal safety and ethical committee of the IIB, UNAM (Protocol Number approval ID 140, approved in 4 April 2018).

### 2.2. Chronic Toluene Exposure

Mice were exposed to 4000 ppm toluene or air (control group) in an inhalation chamber, which consist in a 27 L glass cylinder with a polycarbonate lid. The lid contained an injection port in the outside part and a fan and a wire mesh in the inside part of the chamber. Animals are placed in the bottom of the chamber and an amount of toluene was dropped in the filter paper localized above of the wire mesh, then the fan was turned on. Exposure to toluene or air was conducted 30 min a day for 4 weeks, from Monday to Friday. A photoionization detector (2020 Combo Pro, Inficon, New York, NY, USA) were used to confirm the toluene concentration inside the inhalation chamber. The toluene exposure time and concentration were chosen according to previous studies from our lab, where behavioral and neurochemical alterations have been imitated by repeated toluene exposure [6,9,31,32,33].

### 2.3. Effect of Intranasal Methylprednisolone Administration

Two hours after the last toluene or air (0 ppm toluene) exposure, mice received a single dose of either intranasal isotonic saline solution (IN-SS) or intranasal methylprednisolone (IN-MP) (200 mg/kg). We employed a commercially available formulation of methylprednisolone for intravenous use (methylprednisolone sodium succinate (Pfizer, New York City, NY, USA) and dissolved it with the ampule of injectable water and 2% benzyl alcohol that was provided. Mice received 10 μL in each nostril of the MP solution (200 mg/kg) adjusted to the weight of each mouse. Intranasal doses were applied with a micropipette, placing the tip and slowly releasing the solution into the nostril with the mouse face up in the palm of the hand. Mice were lightly anesthetized with inhaled sevoflurane (10 s) before administration. The dose was ensured to be fully absorbed before the mouse was returned to the cage. Mice were randomly assigned in four experimental groups: toluene + saline solution (TSS), toluene + methylprednisolone (TMP), air + saline solution (ASS), and air + methylprednisolone (AMP). Mice of each group were anesthetized (90 mg of ketamine and 10 mg/kg of xylazine) and perfused with physiological saline followed by 4% paraformaldehyde solution previously cooled to 4 °C, and then euthanized (Figure 1). The expression of ionized calcium binding adaptor molecule (Iba-1) and glial fibrillary acid protein (GFAP) was performed by immunofluorescence analysis in the FC and NA. Hematoxylin and eosin staining were developed to assay histological changes as well as cellular infiltrates in FC and NA. The central level of cytokines (tumor necrosis factor TNF-α, interleukin IL-1β and IL-6) was measured in brain homogenates. Slices of the areas of interest—frontal cortex and nucleus accumbens—were selected according to a histological atlas [34]. Frontal cortex slices were made at 2.34 mm from Bregma, and for the nucleus accumbens, slices were made at 1.42 mm from Bregma. 

### 2.4. Central Inflammatory Mediators

A mixture of ketamine (90 mg/kg) and xylazine (10 mg/kg) was used to anaesthetize all mice. Mice were submandibular bled before being euthanized. Mice were perfused by cardiac puncture with 250 mL of NaCl (0.15 M) to discard the presence of peripheral molecules in central tissues. Brains were removed and processed to determine the protein concentration and cytokine levels (TNF-α, IL-1β and IL-6) and nitric oxide (NO), following the procedure previously described [35]. 

Lysis buffer (50 mM HEPES, 150 mM NaCl, 1% Nonidet-p40, 0.5% sodium deoxycholate, 0.1% sodium dodecyl sulphate (SDS)) containing complete protease inhibitors (Roche, Basel, Switzerland) was employed to homogenate Snap-frozen full brains. Samples were then centrifuged at 16,000× *g* for 15 min at 4 °C and supernatants were collected and kept at −80 °C until analysis. The total amount of proteins in the soluble extract was measured using the Lowry method to normalize the central levels of cytokines [36]. 

### 2.5. Cytokine Enzyme-Linked Immunosorbent Assay (ELISA)

The concentrations of the proinflammatory cytokines IL-1β, IL-6, and TNF-α in brain extracts were measured using commercial kits (BioLegend, San Diego, CA, USA), following the procedures previously reported [37]. ELISAs were performed in 96-well flat-bottomed MaxiSorp microtiter plates (Nunc, Roskilde, Demark). The capture antibody was placed in the microplates for 18 h at 48 °C and then washed with phosphate-buffered saline (PBS)—Tween 20 (0.05%) and blocking for 60 min at room temperature with 2% PBS bovine serum albumin (BSA). Plates were then incubated for 2 h at room temperature for 2 h with standard or samples, washed three times, and incubated at room temperature for 1 h with the detection antibodies at room temperature. Antibodies bounded were revealed using a d 1:1000 dilution of avidin-horseradish peroxidase (HRP) and 3,3′,5,5′-tetramethylbenzidine (TMB). The optical density was read before and after the reaction was stopped with H_2_SO_4_·2N at 450 and 630 nm, respectively, using an DR-200Bc microplate reader (Diatek Instruments, Wuxi, China). Results were expressed in pg/mL per mg of protein in the respective soluble extract. Groups of 4–5 mice for each experimental condition were employed. 

### 2.6. Nitric Oxide Assay

Nitric oxide assay was carried out by indirect measurement of nitrite quantification (NO) using the Griess reaction [38]. The analysis was performed using 96-well MaxiSorp flat-bottom microtiter plates (Nunc, Roskilde, Denmark) where 25 µL of nitrite standards were initially added in duplicate at concentrations between 1.87 and 100 µM/mL followed by 25 µL of the samples in duplicate, then 25 µL of sulfanilamide was added to each well and incubated for 15 min at room temperature with constant shaking, and finally, 25 µL of the NED reagent was added to each well and the absorbance was measured at 10 min at 540 nm. The total amount of proteins in the soluble extract was measured using the Lowry method to normalize the central levels of NO [36]. The results were expressed in µM/mL per mg of protein in the respective soluble extract. Groups of 4–5 mice for each experimental condition were employed. 

### 2.7. Immunofluorescence (IFC) Analysis

Immunohistological studies were performed using 30 µm-thick frozen brain coronal sections that were obtained through cryo-sectioning and preserved in phosphate-buffered saline (PBS) 1× until analysis. After the treatment with citrate buffer (0.01 M citric acid, 0.05% Tween 20, pH 6.0) at 70 °C for 50 min, samples were washed thoroughly with a solution of 2% immunoglobulin (Ig)G-free albumin (Sigma, St Louis, MO, USA) in TBS for 20 min at room temperature. Brain sections were then incubated overnight at 48 °C with either rabbit anti-GFAP (Invitrogen, Carlsbad, CA, USA) or anti-Iba-1 (Wako Chemicals Inc., Richmond, VA, USA) in TBS-2% BSA to detect astrocytes and microglia, respectively. After washing, sections were incubated for 1 h at room temperature with AlexaFluor 594 goat anti-rabbit IgG (Molecular Probes, Eugene, OR, USA) diluted in TBS-2% BSA. Sections were mounted onto glass slides in Vectashield medium (Vector Laboratories, Burlingame, CA, USA) containing 4′,6-diamidino-2-phenylindole (DAPI) to visualize nuclei The mean fluorescence intensity was quantified using Image J software (National Institute of health, Bethesda, MD, USA). Three mice for each experimental condition were analyzed.

### 2.8. Histological Analysis

Hematoxylin-eosin staining was carried out to assay histological changes and cell infiltrates. Briefly, brain sections were fixed with 4% paraformaldehyde for 24 h; subsequently, they were postfixed for 24 h more and embedded in paraffin. Using a microtome, sections with 10 µm thicknesses were obtained from both FC and NA and the slices were mounted onto glass slides. Once the brain sections were obtained, they were deparaffinized and rehydrated on an alcohol gradient, washed with distilled water, and stained with hematoxylin for 10 min. For the eosin staining, the glass slides, initially stained with hematoxylin, were washed in distilled water and treated with 1% acid alcohol and saturated lithium carbonate solution; finally, the slides were placed in an eosin solution for 30 s. Photographs were obtained using a digital camera attached to a light microscope (Nikon DIGITAL SIGHT DS-Ri1, Nikon DIGITAL SIGHT DS-Ri1, Nikon, Tokyo, Japan). 

### 2.9. Analysis of Addictive Behavior by Measuring the Locomotor Activity

To study the addictive behavior induced by chronic toluene inhalation, the locomotor activity of mice in the exposure chamber placed on a surface marked with 4 × 4 cm quadrants was recorded during the last 5 min of exposure, on day 1 and every 7 days during the following 28 days. The total number of quadrants that animals crossed during the test period was recorded. 

Other group of mice that received the same schedule described above were treated on day 28 with saline or with methylprednisolone. The locomotor activity of all mice was measured 24 h after treatments.

### 2.10. Statistical Analysis

All data are expressed as the mean ± standard error. To evaluate the anti-neuroinflammatory effects of IN-MP administration, in animals previously exposed to toluene, ANOVA followed by Tukey test were performed. A difference was considered statistically significant at *p*-value less than 0.05. All analyses were carried out using the Sigma-Stat program (version 3.5, Jandel Scientific, San Rafael, CA, USA).

## 3. Results

### 3.1. Effect of MP on the Central Levels of Cytokines

To evaluate the effect of IN delivery of MP on neuroinflammation induced by toluene exposure, central levels of proinflammatory cytokines were assayed (Figure 2). The exposure to toluene significantly increased the level of TNF-α and IL-6 in brain homogenates and did not affect the level of IL-1β. The administration of MP on the group previously exposed to toluene significantly reduced the levels of TNF-α but not IL-6 in brain homogenates 24 h after the treatment. No effect was observed in the levels of IL-1β. A slight but statistically significant increase in IL-6 levels was observed in the air MP-treated (AMP) group compared with air saline-treated (ASS) group. 

### 3.2. Intranasal Delivery of MP Effectively Reduced Central Levels of Nitric Oxide

The effect of MP on the NO is shown in Figure 3. The results showed that chronic toluene exposure significantly enhanced the NO levels in brain homogenates that were significantly reduced after MP treatment. 

### 3.3. Expression of Iba-1 and GFAP in FC and NA after 24 of Treatment

Two hours after the chronic toluene exposure, mice received IN-SS or IN-MP. Twenty-four hours later, the effect of MP on the activation of microglia and astrocytes was evaluated in the NA and FC. Figure 4A shows an overview of the two regions (frontal cortex and nucleus accumbens) that were evaluated. Figure 4B shows a representative image of the expression of GFAP and Iba-1. As depicted, repeat toluene administration increased the immunoreactivity of GFAP (*p* = 0.09) in NA and FC. Meanwhile, MP treatment decreased the mean fluorescence intensity of GFAP expression, and no changes were observed in the AMP group (Figure 4C). 

Toluene treatment does not significantly modify the expression of Iba-1 (*p* > 0.05) in both areas, in comparison to AMP or ASS groups. In contrast, in the TMP group, the expression of Iba1 was significantly increased compared to the TSS group (*p* < 0.001), probably due to the establishment of a M2 polarized microgliosis (Figure 4C).

### 3.4. Reduced Histological Alterations by Intranasal Methylprednisolone

Hematoxylin-eosin staining was performed to describe histological changes as well as cell infiltration induced by the toluene administration at the level of FC and NA (Figure 5). In the brains of mice that were only exposed to air, there were no abnormalities in the FC or in the NA. In the brain sections of the toluene-administered mice, nuclear pyknosis and cytoplasmic compaction, highly suggestive of apoptosis, were detected. In addition, hydrotropic degeneration, alteration in nuclear contours, and cell rupture—all characteristic cellular processes of necrosis—were found. Similar results were found in the NA (Figure 5). IN administration of MP results in a decrease in brain impairment in both areas. 

### 3.5. Analysis of Addictive Behavior Measuring the Locomotor Activity

Figure 6 shows that exposure to 4000 ppm of toluene produced a significant increase in the locomotor activity from the first week of exposure that is maintained from day 14 to day 28. At day 28, one dose of MP was IN administered to all mice (air and toluene treated). One day later, the locomotor activity was recorded to measure the anti-neuroinflammatory effect on mice behavior. As shown in Figure 6, MP treatment did not modify the locomotor activity of toluene-treated mice. Moreover, MP significantly increased the locomotor of those mice only treated with saline isotonic solution.

## 4. Discussion

Based on the results obtained in the present study, chronic toluene exposure induced marked neuroinflammation characterized by increased levels of NO, TNF-α, and IL-6, as well as increased expression of GFAP. Moreover, histological alterations were observed by the toluene administration. Interestingly, IN-MP not only decreased the central levels of nitric oxide but also the levels of TNF-α. In addition, the expression of GFAP was significantly reduced after the IN-MP treatment. As a result, MP administration was able to reduce histological alterations produced by the chronic toluene treatment.

Clinical and neuroradiological findings sustain the central damage that is caused by repeated use of inhalants such as toluene [39]. The initial injury conceivably results in the expression of damage signals (DAMPS) and activation of the neuroinflammatory response. Once triggered, the sustained neuroinflammation induced by toluene may cause a succession of secondary brain-damage events leading to neuronal damage, such as the enhancement in apoptotic markers [6]. The present results are consistent with the literature on the neuroinflammatory effect of the toluene. Thus, repeated toluene exposure in rats enhances IL-1β in the FC [16]. Previous reports had shown how chronic exposure to toluene activated the NF-Kb signaling pathway through TLR-4 activation mediated by the expression of DAMPs [18]. Moreover, other works also demonstrated that toluene promoted the expression of TNF-α in different brain areas, such as the frontal cortex, hippocampus, and the substantia nigra [33]. Similarly, this study corroborates a previous work where repeated exposure to toluene induces oxidative stress, enhancing nitric oxide and reactive oxygen species production as well as reducing superoxide dismutase activity and glutathione levels in the FC and hippocampus [40]. Astrocytes and microglia are the main cells that respond to toxic damage in the central nervous system during toluene exposure. Experimental models have been used for the study of gliosis caused by toluene. From the present study, repeated doses of 4000 ppm of toluene administration enhance GFAP expression in mouse brains. In line with these results, in a rat study, toluene administration (1500 ppm for 4 h on 4–10 days) augmented GFAP immunoreactivity in the hippocampus and cerebellum [20]. Likewise, the mRNA of neuroimmune markers TNF-α, TLR-4, Iba-1, and GFAP have been found to be increased in the hippocampus of mice previously treated with a low level of toluene [19]. Likewise, toluene 8000 ppm, twice a day for five days increases the GFAP levels in the FC [16]. Altogether, these data corroborate that toluene can induce neuroinflammation. In our study, the chronic administration of toluene was able to increase several inflammatory conditions, such as NO, IL-6, TNF-α, and the expression of GFAP. No increase in IL-1β or Iba-1 could be attained. The production of IL-β requires two signals, one triggered by the recognition of DAMPs that induces the expression of the pro-IL-1β and the second coordinated by the activation of the inflammasome (which activates caspase 8 that cut the pro-IL-1, leading to the mature and active form of this inflammatory cytokine). It is possible that the inflammasome activation could not occur in our model induced by chronic inhalation of toluene. In contrast, in models of toluene with fewer administrations (10 exposures), the activation of NLRP3 inflammasome with the mRNA expression levels of IL-1β has been observed [16]. It is possible that the chronicity in the toluene administrations lead to different outcomes in the production of IL-β.

Herein, we explore the potential of anti-neuroinflammatory treatment using intranasal glucocorticoid administration. MP was selected considering that it is a synthetic glucocorticoid, extensively used orally or parenterally for its anti-inflammatory and immunosuppressive effects [26]. It has been widely administered to control the inflammatory process in several pathological diseases and it is the most common glucocorticoid employed for the treatment of relapses in multiple sclerosis [41,42,43]. However, high doses of MP have been used to reach the required concentration in the central nervous system to control neuroinflammation that may lead severe negative side effects that limit its use [44]. The intranasal (IN) route is a feasible alternative for drug delivery directly to the CNS [45,46,47] through the olfactory and trigeminal nerve structures avoiding the blood–brain barrier. In our research group, we recently demonstrated that intranasally administered glucocorticoids are more effective to control the neuroinflammation that accompanied different diseases, in comparison to intravenously administered ones [37,48]. In the murine model of multiple sclerosis, we found that MP intranasally administered was able to reduce the inflammatory cytokines as well as the expression of activation markers of glia and astrocytes (Iba1 and GFAP, respectively) and improved the clinical score [27]. In this study, the treatment with only one dose of IN-MP significantly reduced some inflammatory features, such as TNF-α, ON, and astrocytes activation, results that are expected when potent anti-inflammatory drugs are employed [49]. Indeed, the damage induced by chronic exposure to toluene is partially reverted as soon as 24 h after IN-MP. This effect may be the result of the rapid reduction in neuroinflammation through the repression of inflammatory transcription factors together with the endothelial repair through the strengthen cell-to-cell contacts [50,51]. However, MP treatment could only partially reduce inflammation by lowering TNF-α and NO levels and maintaining slightly increased levels of the pleiotropic cytokine IL-6. IL-6 is a pro-inflammatory cytokine that is frequently accompanied by different neurological diseases and leads to astrogliosis [52]. However, when the inflammatory process begins to cease, IL-6 behaves as an anti-inflammatory cytokine inhibiting the expression of TNF-α- or IL-1β induced by ICAM-1 expressed in primary astrocytes. Additionally, it has been reported that IL-6 functions as an anti-inflammatory cytokine, helping to maintain the blood–brain barrier integrity in neuroinflammatory conditions. In line with these anti-inflammatory functions, IL-6 also induces the survival, proliferation, differentiation, and regeneration of neurons, influences the synaptic release of neurotransmitters, and promotes astrogliogenesis and oligodendrogenesis [53]. Thus, the high levels of this cytokine in MP-treated mice may reflect the restorative conditions of the inflamed brain. Although a single dose of MP showed benefits reducing some inflammatory markers, it is also likely that more time or additional doses of MP will be required to reverse the values of IL-6. A recent study evaluated minocycline, an anti-inflammatory agent, on the effects induced by toluene at the level of the FC. Similar to our results, minocycline also reduced neuroinflammatory marker IL-1β, while this drug was able to reduce the increased levels of GFAP induced by toluene [16]. A direct comparison is hard to conduct, on the aims of Cruz et al. study was to prevent the inflammatory response induced by toluene, the experiments were conducted in rats, the administration of minocycline was intraperitoneal and the neuroinflammation was evaluated in the FC. In the present study, the objective was to revert the brain damage already induced by toluene, mice were employed, the administration of the steroid was performed intranasally, and the whole brain was used to analyze the proinflammatory cytokines. Despite these differences, both studies show evidence that toluene can induce inflammation at the cerebral level and anti-inflammatory drugs are able to reduce some of those parameters of neuroinflammation. This study tries to conduct an approximation to evaluate the therapeutic potential of MP in the neuroinflammatory process induced by toluene; consequently, the whole brain was employed to analyze some neuroinflammation markers; however, a more specific analysis of the brain area is currently underway in our laboratory.

Exposure to toluene produces marked histological changes by altering the lipid structure of cells, as well as by interacting with different cellular proteins [33]. In patients who have been exposed for more than 10 years, chronic toluene usage leads to demyelination and axonal degeneration [54]. Moreover, there is evidence showing that toluene administration triggers tissue damage due to an increase in oxidative stress as well as the expression of proapoptotic proteins [6,40]. As we hypothesized, the reduction in neuroinflammation was reflected in greater integrity of the neuronal tissues, as observed in the histological profile, where less pyknotic and necrotic cells were observed in the group previously exposed to toluene as shown in Figure 5. The increased Iba-1 expression induced by MP could also contribute to this histological improvement. Indeed, it has been extensively reported that steroids promote the activation of alternative activated microglia, M2 microglia [55]. 

In the present study, chronic exposure to toluene produced behavioral sensitization reflected in a progressive increase in locomotor activity (Figure 6). It is important to highlight that this increase was significant from the first week of exposure (*p* < 0.001), which allowed establishing the rapid effect of toluene to induce these changes. Our results coincide with those reported by other authors that found in a rat model that chronic exposure to toluene for 12 days for half an hour significantly increased locomotor activity. In our study, the administration of MP had no effect in reversing the behavioral sensitization induced by toluene. Moreover, MP administered to mice that only received air significantly increased their locomotion. These results are compatible with two aspects: on the one hand, the control of neuroinflammation does not reverse the addictive behavior at least in a short time. Additionally, the increased movements associated with the administration of the glucocorticoid MP may be due to changes in the mice metabolism resembling the result of the “corticoid-euphoria” reported in humans after glucocorticoids treatment [56]. Our results disagree with other reports that suggest that neuroinflammation caused by cell damage due to drug abuse could reinforce addictive behaviors [57,58]. Further research using different glucocorticoids schedules will be necessary to establish the relationship between neuroinflammation and addiction behavior.

Overall, the results obtained in this study showed that MP mitigated the proinflammatory markers and brain damage induced by the exposure to toluene, which allows us to propose the interest of evaluating other glucocorticoids and different administration regimens for the control of secondary damage induced by sustained neuroinflammation induced by chronic consumption of toluene, which could impact the quality of life of inhalant users.

## Figures and Tables

**Figure 1 pharmaceutics-14-01195-f001:**
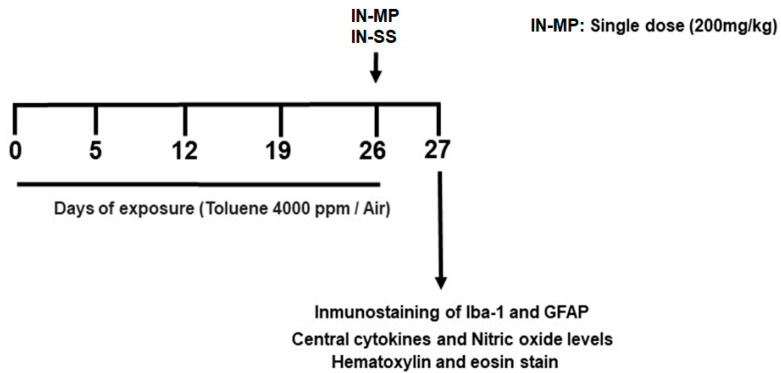
Experimental design. At the beginning animals were exposed to toluene 4000 ppm or air for four weeks. Two hours after the last exposure the animals received a single dose of intranasal MP (IN-MP) or intranasal SS (IN-SS). One day later, the expressions of Iba-1 and GFAP were assayed by immunofluorescence and central cytokines and nitric oxide levels were evaluated by enzyme-linked immunosorbent assay (ELISA) and Griess reaction, respectively. Histological analysis was performed with hematoxylin and eosin staining 24 h after the treatment.

**Figure 2 pharmaceutics-14-01195-f002:**
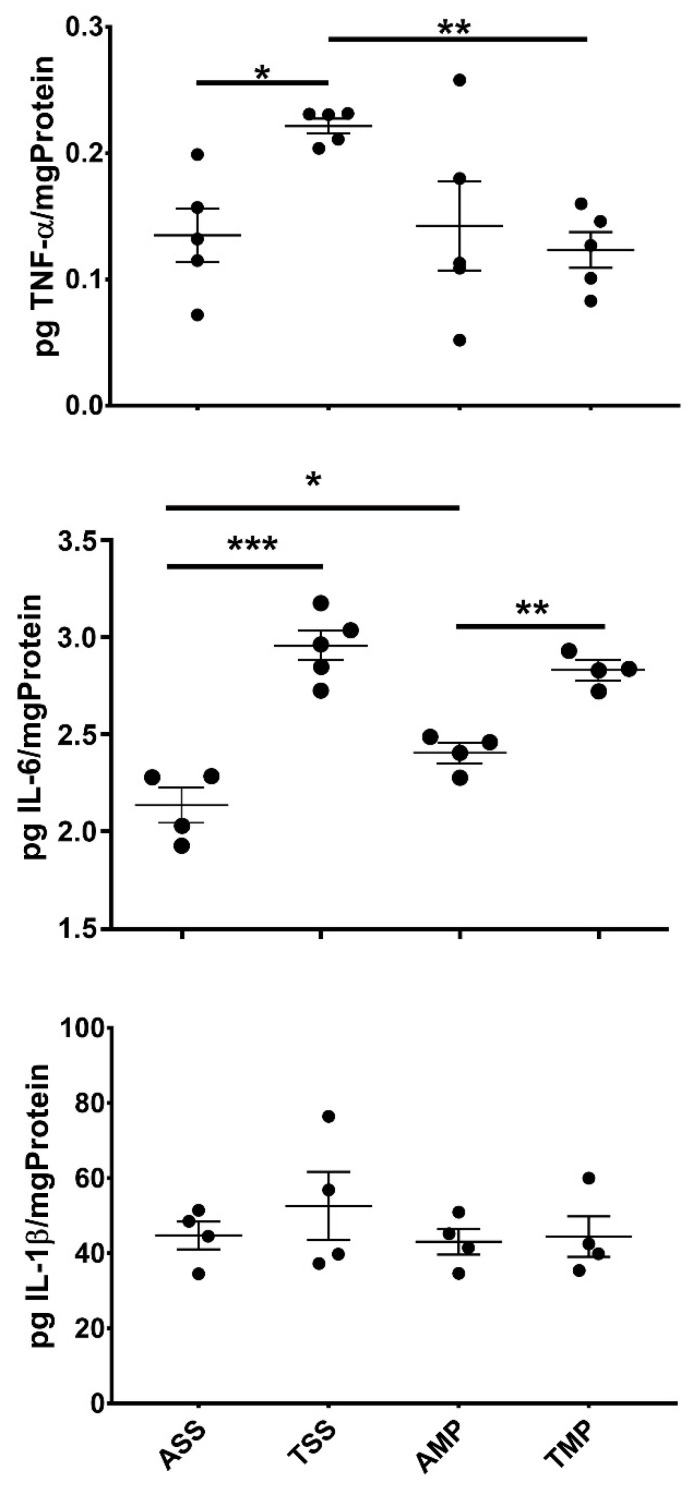
Central cytokine levels (pg/mg protein) in soluble extract from brain of mice chronically exposed to toluene and analyzed by ELISAs test. All results are showed as the mean ± standard error of groups values. * *p* < 0.05, ** *p* < 0.01, *** *p* < 0.001. ANOVA followed by Tukey’s test was performed. Air + saline solution (ASS); toluene + saline solution (TSS); air + methylprednisolone (AMP); toluene + methylprednisolone (TMP).

**Figure 3 pharmaceutics-14-01195-f003:**
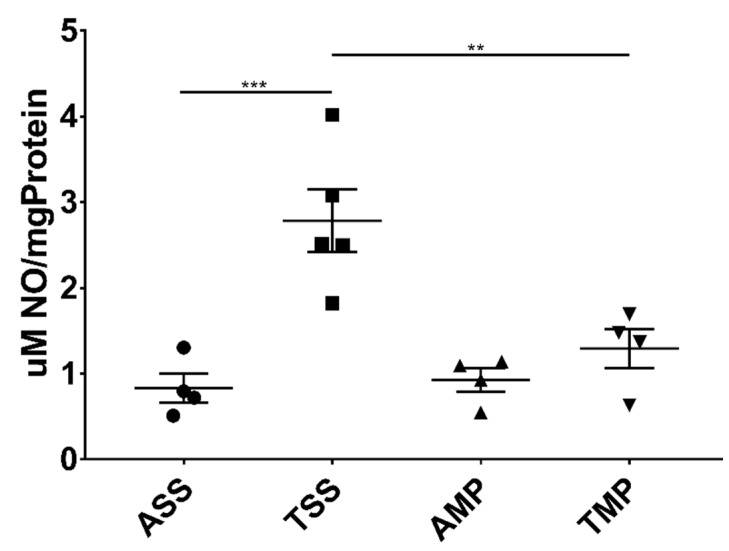
Central nitric oxide levels (µM/mg protein) in soluble extract from brain of mice repeatedly exposed to toluene and measured by Griess reaction. All results are shown as the mean ± standard error of groups values. ** *p* < 0.01, *** *p* < 0.001. ANOVA followed by Tukey’s test were performed. Air + saline solution (ASS); toluene + saline solution (TSS); air + methylprednisolone (AMP); toluene + methylprednisolone (TMP).

**Figure 4 pharmaceutics-14-01195-f004:**
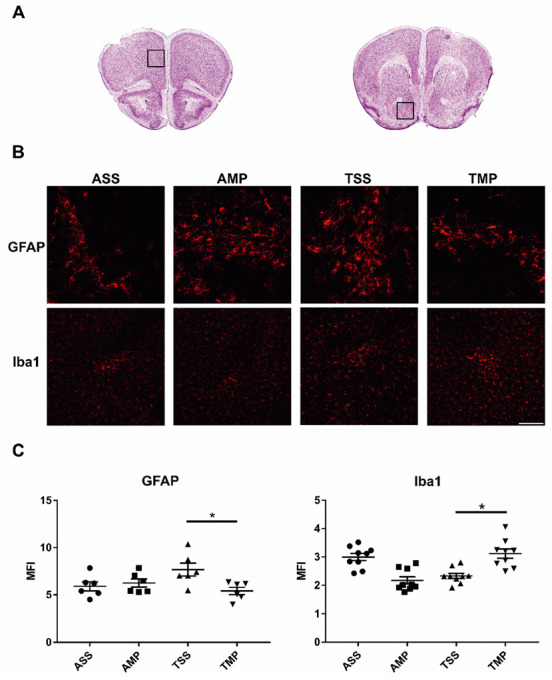
Analysis of brain GFAP and Iba-1 expression in exposed to toluene and non-exposed to toluene mice 24 h after the IN-MP administration. (**A**) Histological representation of analyzed areas. Squares points out areas in frontal cortex (**left**) and nucleus accumbens (**right**) which were analyzed. Images taken from “The mouse brain in stereotaxic coordinates” Reproduced from [30], OXFORD UNIV PRESS INC, 2020. (**B**) Representative images of GFAP and Iba-1 immunofluorescence stains. Scale bar represents 100 µm. (**C**) The mean fluorescence intensity (MFI) of GFAP and Iba-1 in frontal cortex and nucleus accumbens were quantified using the Image J software. All results are showed as the mean ± standard error of groups values. * *p* < 0.05. ANOVA followed by Tukey’s test. Air + saline solution (ASS); air + methylprednisolone (AMP); toluene + saline solution (TSS); toluene + methylprednisolone (TMP).

**Figure 5 pharmaceutics-14-01195-f005:**
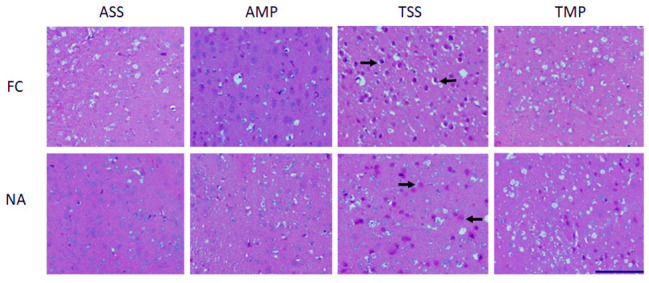
Histological examination of brains of mice repeatedly exposed to toluene and those with a previous exposure toluene and then treated with methylprednisolone. Hematoxylin-eosin stains of brain sections at frontal cortex (FC) and nucleus accumbens (NA). Scale bar represents 100 µm. Air + saline solution (ASS); air + methylprednisolone (AMP); toluene + saline solution (TSS); toluene + methylprednisolone (TMP). Arrows show nuclear damage in TSS group in both FC and NA regions. Treatment with IN-MP reduced nuclear damage (TMP).

**Figure 6 pharmaceutics-14-01195-f006:**
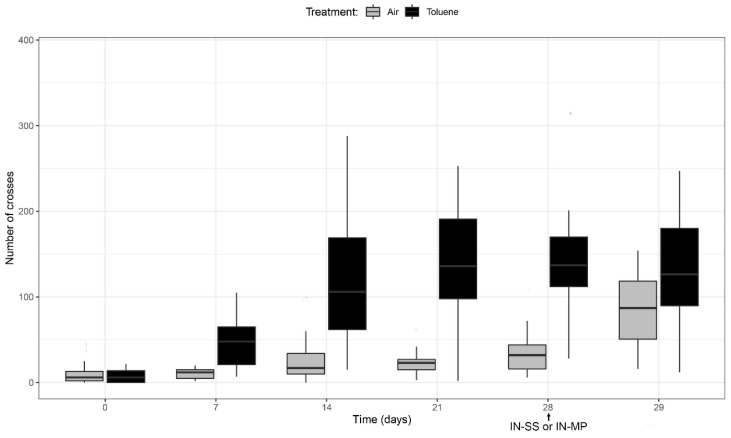
Mean ± SD of the number of crosses performed by 68 mice, half treated with air and the other half with toluene for 28 days. A group of 30 mice were then treated with one dose of IN-MP (200 mg/kg) and the locomotor activity was registered 24 h later. Values were compared by a T-student test.

## Data Availability

Data available at dx.doi.org/10.6084/m9.figshare.18343253 (accessed on 9 November 2021).

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
