# Peer review of "Intranasal Methylprednisolone Ameliorates Neuroinflammation Induced by Chronic Toluene Exposure"

_pharmaceutics, 2022, doi:10.3390/pharmaceutics14061195_

Round 1

Reviewer 1 Report

The authors described the effects of toluene inhalation on brain inflammation, which can be ameliorated by methylprednisolone. The results contained some interesting points. However, the results are preliminary and lack of mechanistic insights. Moreover, insufficient experiments make the conclusion less convincing. Following are some of the major points to be considered.

1、The novelty and content of this study needed to be explained. Previous studies showed many drugs had similar effects in this model. Please explain the purpose of this study and add more clinical relevance in the introduction or discussion section.

2、MP treatment group was able to increase IL-6 expression. Could this drug also induce inflammatory response? More evidences are needed to exclude this possibility.

3、The conclusions were not well supported by the amount of data. For example, the authors claimed that toluene inhalation could cause the release of inflammatory factors by enhancing TLR4 and other inflammatory signaling pathways, but did not check this signaling pathway. In addition, the author only tested nitric oxide as the products of oxidative stress, which make the claims doubtful.

4、Explain in detail the different responses of the two glial cells to drugs shown in figure 4 in result part.

Reviewer 2 Report

The manuscript with the title “Intranasal methylprednisolone ameliorate neuroinflammation induced by chronic tolu-ene exposure” describes the evaluation the anti-inflammatory effect of a single dose of intranasal methylprednisolone in a mouse model of chronic exposure to toluene.

From the introduction, it was not clear what is the context/opportunity for the treatment of people abusing form inhalant substances, or whether any treatment is used and which limitations it may have. Are corticosteroids used at all? (I understood not, please state it clearly). What is the unmet need targeted here? Then, in the conclusion, the interpretation is that the results suggest the interest of evaluating other glucocorticoids and different administration regimens for the control of secondary damage induced by sustained neuroinflammation induced by chronic consumption of toluene… but how are these patients identified? They come to the hospital due to complications? Are followed in some way? Could be treated when following treatments to the addiction? (corticosteroids cannot be given preventively to young people at risk of consumption). This makes it difficult to evaluate the impact/relevance of the work from an applied point of view.

There is no description of how methylprednisolone was formulated for intranasal administration or of how the administration was performed (in which volume/technique), and the dose was quite high (200 mg/Kg), which might be difficult to administer intranasally. Furthermore, the discussion in the introduction of the possible advantages of the intranasal route may be misliding, sice that in the presented work there is no demonstration that the dose can be reduced compared to parenteral administration, or that systemic exposure is reduced. If a large volume od a drug suspension is being administered to mice (which can be suspected form the large dose), it is expected that absorption is occurring also from the gastrointestinal tract and/or from the lungs. Therefore, the relevance of the intranasal route as the administration route is unknown. There is no problem in having used intranasal administration… well, no clear advantage either, since drug absorption likely did not occur within the nasal cavilty in this case. Anyway, the intranasal route is not stressed (and well) in the discussion and conclusion.

Despite the uncertain local of drug absorption, the treatment does seem to revert some of the alterations induced by toluene, and may be if interest if it is indeed the first time corticosteroids are tested to counteract toluene induced neuroinflammation. If not, these results should be discussed/compared to other similar treatments. It is referred that “A recent study evaluated minocycline, an anti-inflammatory agent, on the effects induced by toluene at the level of the FC.” Minocycline is primarily known as an antibiotic (class of tetracyclines)… maybe it would be preferable to say something like “A recent study evaluated the anti-inflammatory effect of minocycline, a tetracycline antibiotic, on the effects induced by toluene at the level of the FC.”

Importantly, imagens of hematoxylin-eosin stains are very strange looking…. Where does the green come from? These should be replaced by better quality images. And how is the damage accumulated over several days reverted only in 24h? Could you please discuss that? Are the markers under evaluation transient in time and similar to what would be observed form a single exposure to toluene?

English: Generally, the text reads well. Verb conjugation should be revised (s is missing even in the title in “ameliorates”). Some expressions are less common…eg, I believe that “Impairment” is not the same than abnormality and that “histological impairments” is a strange expression, since impairment is usually applied to a function…. There may be histological signs of impairment though.

Finally, I get the impression that the work is somewhat preliminary, and the authors also comment on that, saying it is an exploratory study, by suggesting that additional doses of the drug might revert IL-6 levels but did not test it, and by saying that “This study tries to conduct a first approximation” and that “a most specific analysis by brain area is currently matter of study in our laboratory”. Maybe you could add those results?

Reviewer 3 Report

This is a well-organized and well-illustrated paper, has an important clinical message, and should be of great interest to the readers. This paper evaluated the anti-inflammation effects of intranasally administered methylprednisolone in a toluene induced neuroinflammation, in an animal model. This research opens up new possibilities in diseases associated with neuroinflammation. This manuscript deserves publication after addressing the issues cited below.

  1. I sugges the authors to add a brief sentence in the introduction about the toluene induced neuroinflammation model and other previous works that have employed this model for studying neuroinflammation and how this model is comparable to the clinical neuroinflammation  conditions.
  2. Did the authors evaluate the body weight changes and other behavior changes during the drug administration.
  3. GFAP and IBA-1 staining and Histology imahes in figure 5 are not clear. I suggest the authors to use high quality images for these figures.
  4. In the discussion section i suggest the authors to add previous studies that reported the anti-inflammatory action of methylprednisolone.
